# DA-Transfer: A Transfer Method for Malicious Network Traffic Classification with Small Sample Problem

**Ruonan Wang** [1]**, Jinlong Fei** [1,*]**, Min Zhao** [2]**, RongKai Zhang** [1]**, MaoHua Guo** [1]**, Xue Li** [1] **and Zan Qi** [1]

[1] State Key Laboratory of Mathematical Engineering and Advanced Computing, PLA Strategic Support Force Information Engineering University, Zhengzhou 450001, China

[2] College of Electronic Information and Automation, Tianjin University of Science and Technology, Tianjin 300457, China

[*] Correspondence: feijinlong@126.com

**Abstract:** Deep learning is successful in providing adequate classification results in the field of traffic classification due to its ability to characterize features. However, malicious traffic captures insufficient data and identity tags, which makes it difficult to reach the data volume required to drive deep learning. The problem of classifying small-sample malicious traffic has gradually become a research hotspot. This paper proposes a small-sample malicious traffic classification method based on deep transfer learning. The proposed DA-Transfer method significantly improves the accuracy and efficiency of the small-sample malicious traffic classification model by integrating both data and model transfer adaptive modules. The data adaptation module promotes the consistency of the distribution between the source and target datasets, which improves the classification performance by adaptive training of the prior model. In addition, the model transfer adaptive module recommends the transfer network structure parameters, which effectively improves the network training efficiency. Experiments show that the average classification accuracy of the DA-Transfer method reaches 93.01% on a small-sample dataset with less than 200 packets per class. The training efficiency of the DA-Transfer model is improved by 20.02% compared to traditional transfer methods.

**Keywords:** malicious traffic; small samples; transfer learning; adaptive

## 1. Introduction

The advent of the Big Data era has been accompanied by an explosion of network traffic, and malicious programs and malicious traffic have been generated. Due to the fact that network traffic classification can associate traffic with its generation process, it is often used as the first step in the network malicious resource detection task in the network security domain [1]. Therefore, researchers have explored various methods to solve the network traffic classification problem, and the problem of accurate classification of network traffic has become a hotspot in the field of research.

The traditional traffic classification methods are broadly classified into three main categories [2]: port-based methods [3], payload inspection techniques [4], and machine learning-based methods [5,6]. As network technology evolves, all these methods face technical bottlenecks. For example, the widespread use of random ports and port masquerading has led to a significant decrease in the accuracy of the port-based methods. The payload inspection techniques also make it difficult to obtain packets due to the widespread use of encryption. The machine learning approach relies on human experience to design network traffic characteristics, and its immediacy and responsiveness make it difficult to meet the rapid iteration of the application software.

Deep learning has automated the end-to-end learning capabilities and characterization of high-quality features on raw data. It is therefore widely used in many applications, such as computer vision [6,7], target detection [8,9], natural language processing [10,11],

and in the medical field [12,13]. In recent years, researchers also used deep learning for traffic classification tasks. However, deep neural networks have huge training parameters and rely on a high volume of data, requiring a massive amount of labeled samples. Obtaining labeled data in real-world scenarios is difficult and expensive, especially for malicious traffic samples. Therefore, the classification of small-sample network traffic data is emerging as a new challenge.

Deep transfer learning usually uses a model that is pre-trained on a large dataset as the feature extractor of the target dataset, and then fine-tunes the network by freezing part of the network layer to reduce the parameters that require training to adapt to small-sample datasets. This can solve the small sample problem to some extent. However, a study [14] revealed that the difference in distribution between the source and target datasets in transfer learning affects the network performance, and two datasets with a large difference may lead the target dataset to fail to converge on the transfer network. In addition, researchers usually recommend model transfer freezing parameters based on their own experience or layer-by-layer traversal, which is time-consuming and not general enough. The optimization of the parameter selection for transfer networks can further improve the accuracy and efficiency of classification networks. Therefore, finding the number of frozen layers of the source network structure suitable for the target dataset and the difficulties in fine-tuning the model due to the variability between large datasets and the target dataset have become obstacles for deep transfer learning to solve small-sample classification problems.

This paper proposes a deep transfer learning method with double adaptive transfer to improve the performance of the small-sample malicious traffic classification. The double adaptation includes data adaptation and model adaptation (DA-Transfer). Both data and model transfer adaptation modules jointly improve the accuracy and efficiency of the model. The main structure of the data adaptation module is a neural mapping network, which drives the distribution of the mapped target dataset close to the source dataset, and the mapped target dataset can be better adapted to the transfer network. The transfer adaptive module automatically outputs the network structure parameters, especially recommending the freeze layer strategy of the network structure. The module has the ability of fast recommendation of freezing parameters, which greatly improves the efficiency of model transfer. The two-dimensional adaptive module optimizes the transfer network performance and alleviates the contradiction between the small-sample dataset and the large number of neural network parameters.

The main contributions of this work are as follows.

- We propose a fused deep transfer learning approach, which combines data adaptation and model transfer adaptation to improve the performance of a small-sample classification model for malicious traffic.
- The data adaptation module reduces the distribution distance between the source and target datasets, which improves the transfer network's ability to adapt to new data and enhances the classification accuracy of the transfer model.
- A recommendation method for adaptive network parameters is investigated, which automatically outputs important parameters for transfer networks and improves the efficiency by 20.01% compared to the artificially designed network.

## 2. Related Work

Deep learning-based traffic classification: The traffic classification methods based on deep learning have been a hot research topic as deep learning has made progress in several fields. In an earlier study, Wang et al. [15] proposed a traffic classification method using convolutional neural networks for representation of learning, which converts raw traffic data into images for the automatic extraction of traffic features. The method was validated by using three classifiers in two scenarios, and the results showed that the method can satisfy the accuracy requirements in practical applications. In addition,

Lotfollahi et al. [16] attempted to compare the effectiveness of the stacked auto-encoder (SAE) and convolutional neural network (CNN) in the traffic classification problem. The test results showed that the performance of the CNN was the best when used as the classification model, and the method obtained an average accuracy of 0.94 in the traffic classification task. In terms of the network fusion, Zou et al. [17] proposed a deep neural network by combining CNN and RNN (called CNN + LSTM) and extracted packet-level and stream-level features (i.e., time-series features) to improve the classification accuracy. The experimental results on the ISCX VPN-nonVPN dataset [18] achieved an average accuracy of 0.91. Recently, Cui et al. [19] proposed a novel session-packets-based encrypted network traffic classification model using capsule neural networks (CapsNet), called SPCaps. SPCaps introduces a twice-segmentation mechanism to dilute the interference traffic and increase the weight of effective traffic. Then, it learns the spatial characteristics of encrypted traffic using CapsNet and outputs the results of encrypted traffic classification by a softmax classifier. The model is superior to the most advanced encrypted traffic classification method and achieves a recall rate of more than 0.99 for both application classification tasks and traffic classification tasks on the ISCX VPN-nonVPN dataset. The above research [20–22] has been applied to a variety of network traffic task scenarios and has important reference values for the network traffic problem.

Deep Transfer Learning: Yosinski et al. [23] were the first to investigate the transferability of deep networks; they transferred the network by directly transferring the parameters of the trained network A to network B layer by layer, freezing the parameters of the module transfer, and fine-tuning the remaining parameters of network B. The results of the study found that direct transferring of the parameters of the shallow part of the trained model has less impact on the accuracy of the target model, which confirms the transferability of deep networks. Neyshabur et al. [24] came to a similar conclusion that the lower layers of neural networks typically extract generic features, and the higher layers extract features that are strongly relevant to the task. Furthermore, researchers [25] found that the similarity of the domains plays an important role in capping the performance of transfer learning, i.e., the more similar the datasets are to each other, the better the transfer is. Based on the above research [26,27], it can be concluded that the transfer network greatly reduces the network parameters that need to be trained, which makes the datasets of small-sample tasks match it.

Small-sample traffic classification based on deep transfer learning: Once the transferability of deep networks was proven, researchers started to apply deep transfer learning to the task of classifying small samples of network traffic. Guan et al. [28] applied deep transfer learning to solve the network traffic classification task of scarce datasets in 5G IoT systems. They trained the classification model through the network parameter transfer and network fine-tuning, and achieved adequate classification results. This work confirms that deep transfer learning is effective in solving the classification task of small-sample traffic datasets. In another study, Dhillon et al. [29] applied deep transfer learning to a hybrid CNN-LSTM model, and it performed well in small-sample intrusion detection tasks. Idriss [30] et al. applied the transfer learning method to the intrusion detection system and updated the solution of the intrusion detection system based on deep learning (DL-IDS). This method achieved adequate results on multiple indicators, such as detection rate. Eva et al. [31] proposed a deep learning network based on deep transfer learning to solve the problem of zero-day attack detection. Experiments show that the detection rate of this method exceeds any previous intrusion detection system based on deep learning. The proposed deep transfer learning technology makes it possible to construct a large-scale deep learning model to perform network classification tasks. These models can be deployed in the target domain of the real world, and they can maintain classification performance and improve classification speed even when resources are limited.

## 3. Methods

*3.1. Datasets and Pre-processing Methods*

3.1.1. Dataset

In this paper, two datasets are used to train the proposed transfer network model. One is a large dataset (source dataset) used to train the initial network, and the other is a small-sample dataset (target dataset) used to fine-tune the transferred model and perform classification tests. The details of the two datasets are as follows:

Source dataset: The ISCX VPN-nonVPN traffic dataset [18] consists of captured traffic generated by different applications. In this dataset, the captured packets are divided into different pcap files, whose labels are divided into applications (e.g., Email, SFTP, etc.) and specific activities (e.g., voice calls, chats, file transfers, video calls, etc.). In this paper, we mainly use the application data as the source dataset to train the network. In addition, data balancing is performed on the dataset in order to test the transfer and classification capabilities of the proposed method. Table 1 presents the information of the data in the final dataset.

**Table 1.** Source dataset details.

| Source Data | Highest Visible Protocol | Size (K) | Quantity |
|---|---|---|---|
| Email | SSL&HTTPS | 15,974 | 84,585 |
| YouTube | HTTPS | 623,616 | 271,593 |
| FTPS | HTTPS | 924,600 | 250,652 |
| Vimeo | HTTPS | 830,464 | 367,282 |
| Spotify | HTTPS | 172,032 | 98,600 |
| Torrent | HTTPS | 366,592 | 269,115 |
| Netflix | HTTPS | 1,355,776 | 160,789 |
| SCP | HTTPS | 545,792 | 183,286 |
| SFTP | HTTPS | 1,458,176 | 324,424 |
| Mean | - | 699,224 | 223,369 |

Target dataset: In order to restore the model performance under realistic scenarios, the target dataset captures the attack behaviors of nine Trojans and uses Wireshark to obtain the traffic packets in the process. In order to approach the size of a small-sample dataset under realistic conditions, the target dataset uses part of the captured malicious traffic data. Table 2 presents detailed information and the number of packets of the nine types of captured Trojan traffic. This dataset supported a malicious traffic identification track for a domestic artificial intelligence competition as a public dataset.

**Table 2.** Target dataset details.

| Target Data | Highest Visible Protocol | Size (K) | Quantity |
|---|---|---|---|
| Finalfantasy | SSH | 3180 | 116 |
| Freerat | SSH | 3114 | 200 |
| Chongqinghack | TCP | 1360 | 200 |
| Irat | SSH | 2343 | 175 |
| Poison-ivy | SSH | 2344 | 169 |
| Greydove | HTTP | 2257 | 189 |
| Shangxing2009 | TCP | 2344 | 200 |
| Suncontrol | TCP | 2125 | 200 |
| Ximencontrol | TCP | 2051 | 200 |
| Mean | - | 2346 | 183 |

### 3.1.2. Pre-Processing

The pre-processing implements the filtering and unified format of the network data, including filtering redundancy, truncating data, removing bad samples, and normalization. Firstly, the information in the Ethernet header, which is not useful for traffic classification, is removed. Secondly, the Transmission Control Protocol (TCP) and User Datagram Protocol (UDP) differ in the input format due to different header lengths by injecting zeros at the end of the UDP segment header and making it equal to the length of the TCP header. In addition, the IP address information is masked. The datasets are captured in a realistic simulation environment, discarding meaningless packets that do not contain any payload. Finally, to unify the sample size of the neural network input, a vector with a sample size greater than 1500 is truncated and zeros are filled for byte vectors smaller than 1500. The normalization phase divides each element by 255 to normalize the byte vector.

### 3.2. DA-Transfer Method

### 3.2.1. Overall Framework of DA-Transfer

The overall flow diagram of the proposed method is shown in Figure 1. The DA-Transfer incorporates a data adaptation module and a model transfer adaptation module to improve the classification performance of small-sample traffic. The implementation process of the method consists of three parts: data adaptation, model parameter adaptation, and transfer network training. Firstly, the target dataset is fed into the data adaptation module to reduce the distribution distance from the source dataset to be close to the original attributes of the trained large network. Secondly, the mapped target dataset is input into the model parameter adaptation module together with the source dataset to obtain the recommended freezable network parameters. Finally, the transfer network parameters are adjusted according to the recommended information, and the mapped target dataset is fed into the pre-trained network for fine-tuning to finally realize a small-sample traffic classification model adapted to the target dataset.

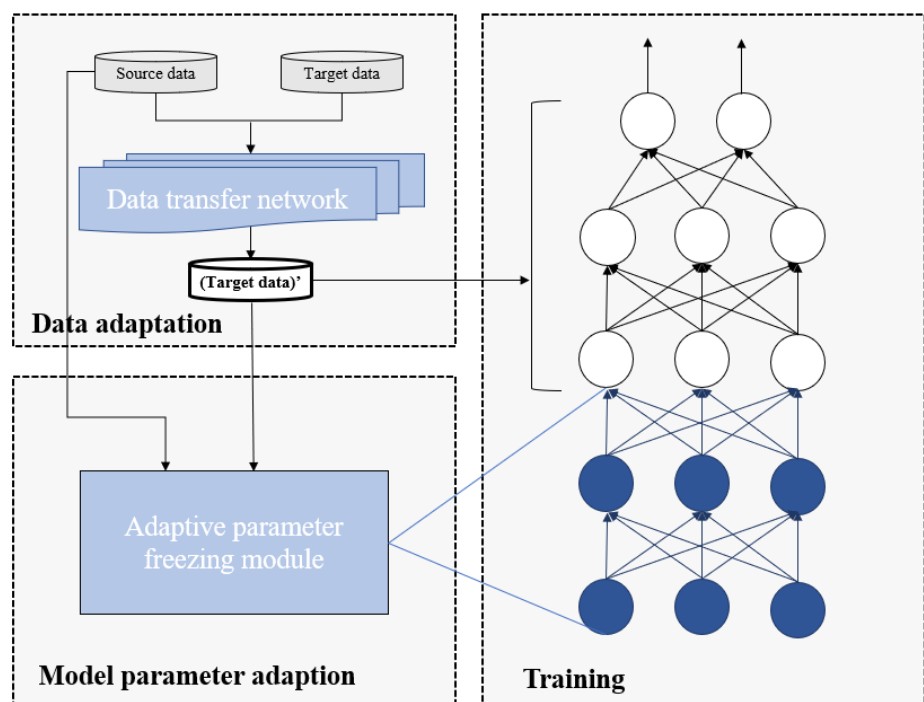

**Figure 1.** Overall framework of DA-Transfer.

### 3.2.2. Data Adaptation

The data adaptation module uses a data mapping network to map the target dataset *Y* to dataset *Y* with a similar distribution to the source dataset *X*. This helps to reduce the range of parameter adjustment in the subsequent fine-tuning process and achieve the purpose of improving the classification accuracy.

The data mapping network is mainly a deep feed-forward architecture, the framework of which is illustrated in Figure 2. Firstly, the source domain datasets are input into three continuous convolutional layers and a fully connected layer neural network to obtain a preliminary mapping of the source domain data. After that, the learning phase of the neural network is entered to make the data distribution distances of *X* and *Y* close to each other, for which we need to define the data distribution distance of the two datasets. The maximum mean difference (*MMD*) [32] is used to quantify the distance between the *X* and *Y* distributions. *MMD* is a nonparametric measure used to compute the distance between distributions based on kernel embedding in the reproducing kernel Hilbert space. Given the domain samples (source) and (target) from two distributions, the *MMD* distance is calculated as follows:

$$MMD(X_S, Y_t) = \left\| \frac{1}{n_s} \sum_{i=1}^{n_s} \varphi(x_i^s) - \frac{1}{n_t} \sum_{i=1}^{n_t} \varphi(y_i^t) \right\|_\wp \tag{1}$$

where $\varphi(x)$ maps each instance to the kernel $k(x_i, x_j) = \phi(x_i)^T \phi(x_j)$ related Hilbert space $\wp$, and $n_s$ and $n_t$ are the sample sizes of the source and target domains. By minimizing the *MMD* distance as the goal of the neural network training, convergence is reached after multiple rounds of network training to obtain the data mapper.

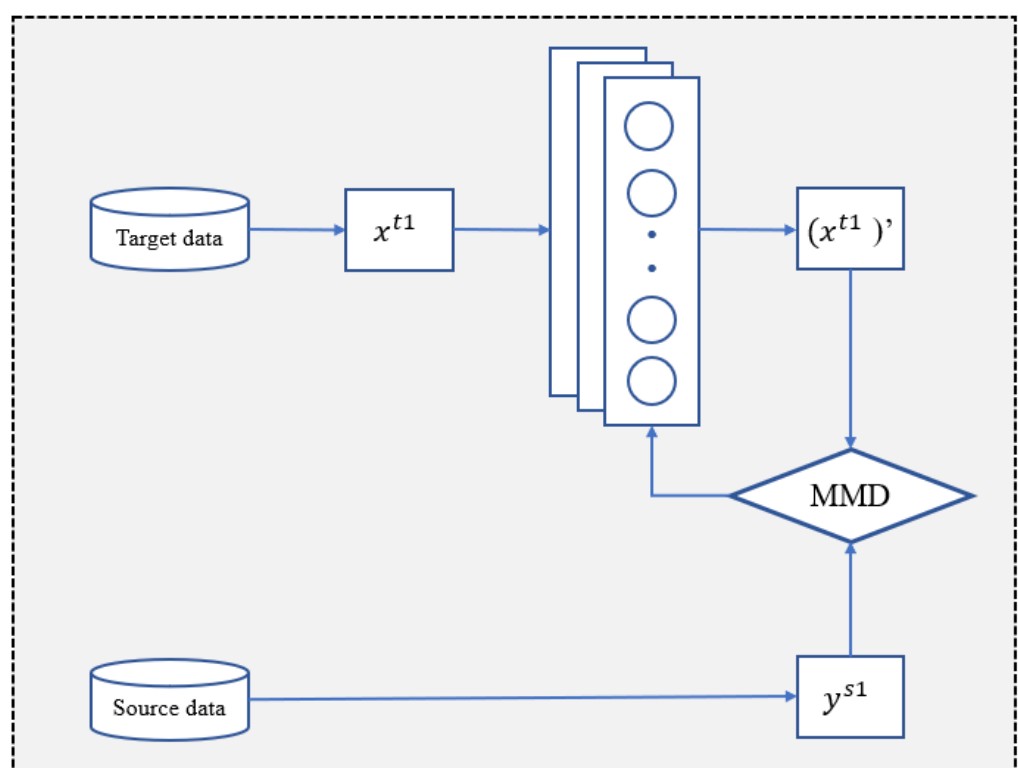

**Figure 2.** Data adaptation mapping network.

### 3.2.3. Model Transfer Adaptive

When transferring a model from a source dataset to a target dataset, filtering the number of layers of network parameters that the source and target datasets can share

ensures that the transfer model retains its ability to extract common features. The traditional approaches are mostly based on experience with the dataset or by freezing the network parameters layer by layer and fine-tuning the parameters of the remaining layers [33]. Finding the optimal number of network layers, with shareable parameters based on the final classification accuracy, is effective but time-consuming. The goal of this work is to efficiently and accurately derive the number of layers so that the network can freeze parameters.

The principle of freezing the parameters of a network layer is that the features extracted in that layer are common to the source and target datasets, i.e., the features extracted in that layer cannot identify whether the input data belong to the source dataset or the target dataset. Thus, the problem can be transformed into finding the number of network layers with higher losses when performing the source dataset and target dataset binary classification task. Assume that $x \in (X, Y)$, where $X$ and $Y$ are the source and target datasets, y is the label for x, and $y \in \{0,1\}$. The detailed steps are as follows.

Step 1: Input x into the first n layers of the pre-trained neural network, and save the feature vectors' output by the first n layers of the neural network.

Step 2: The feature vectors of the first n layers are input into the fully connected layer. The binary classification task of determining whether x comes from the source dataset or target dataset is used as the goal, and the loss values of the model output with label y are calculated and saved.

Step 3: Calculate the gradient of the loss value decline (shape of the loss curve is shown in Figure 3). When the gradient increases significantly, freeze the network layer parameters.

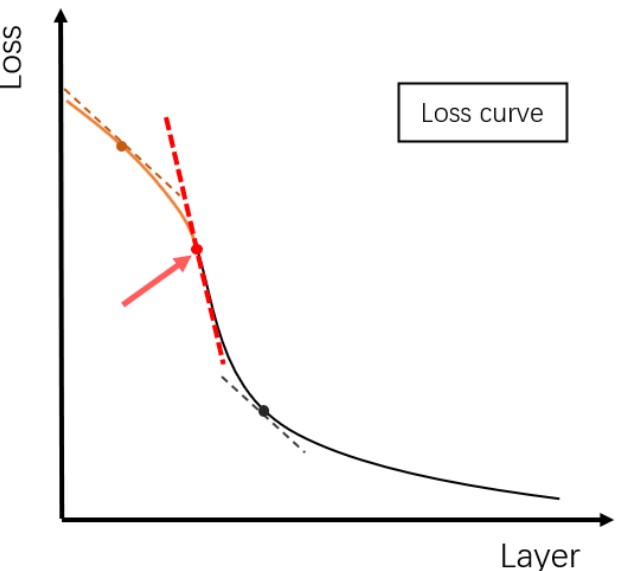

**Figure 3.** A rapid change in the gradient occurs near the loss curve of the appropriate frozen layer number.

### 3.2.4. Model Architecture and Training

We built a source network based on 1D-CNN. It includes five one-dimensional convolution layers, three maximum pooling layers, three fully connected layers, and one classification layer. The number of neurons in the convolutional layer is set to 200, and a ReLU activation function is added after each convolutional layer. The two-dimensional tensor output by the convolutional layer is compressed into one dimension and sent to three consecutive fully connected layers. To prevent overfitting, a dropout rate of 0.05 is taken

after each fully connected layer, so a random set of neurons is set to 0 in each iteration. The detailed network hyperparameters are shown in Table 3.

The source network is trained on the large dataset ISCX VPN-nonVPN, and then fine-tuned on the target dataset as a transfer network. We set the batch size to 64, epoch to 20 rounds, and the initial learning rate to 0.01. When the model is fine-tuned on the target dataset, the learning rate is set to 0.001 because the model has converged on the large dataset.

The experiment optimizes the network hyperparameters from several angles. Firstly, the network convolution layer is set to five layers, the maximum pooling layer is set to three layers, the linear layer is set to four layers, and the filter is set to 200 because of the network complexity that cannot be supported by a small sample size. Secondly, the transfer adaptive module uses a self-search method to automatically optimize the recommended freezing parameters. In addition, we refer to the empirical parameter settings of the current network: the initial learning rate is set to 0.01, the convolution kernel size is set to 5, and the dropout is 0.05.

**Table 3.** Network hyperparameters.

| Operation | Kernel Size | Strides | Channels | Dropout | Nonlinearity |
|---|---|---|---|---|---|
| Input packet | - | - | - | - | - |
| Convolution | 5 | 1 | 200 | 0.05 | ReLU |
| Convolution | 5 | 1 | 200 | 0.05 | ReLU |
| Max pooling | - | 2 | - | 0.05 | - |
| Convolution | 5 | 1 | 200 | 0.05 | ReLU |
| Convolution | 5 | 1 | 200 | 0.05 | ReLU |
| Max pooling | - | - | - | 0.05 | - |
| Convolution | 5 | 2 | 200 | 0.05 | ReLU |
| Max pooling | - | - | - | 0.05 | - |
| Flattening | - | - | - | - | - |
| FC | - | - | 200 | 0.05 | ReLU |
| FC | - | - | 100 | 0.05 | ReLU |
| FC | - | - | 50 | 0.05 | ReLU |
| Output FC | - | - | 9 | - | Softmax |

## 4. Results

### 4.1. Classification Performance

#### 4.1.1. Malicious Traffic Dataset Testing

We randomly divide the target dataset into three separate sets, where 50% of the samples are used for training and adjusting weights and biases, 30% of the samples are used for validation in the training phase, and the last 20% of the samples are used for testing the model. In order to better evaluate the classification performance of the DA-Transfer model, we introduce the confusion matrix, as shown in Table 4. This matrix

describes the number of samples in the dataset that are correctly or incorrectly classified by the classifier, and it is commonly used in classification problems.

**Table 4.** Confusion matrix.

|  |  | Predict | |
|---|---|---|---|
|  |  | Positive | Negative |
| Actual | Positive | True Positive (TP) | False Negative (FN) |
|  | Negative | False Positive (FP) | True Negative (TN) |

Based on the confusion matrix, three metrics are often used to evaluate the model. *Precision* refers to the proportion of instances that are predicted to be correct and positive in all the instances. It is expressed as follows:

$$Precision = \frac{TP}{TP + FP} \tag{2}$$

The abnormal class *recall* rate refers to the proportion of all the positive classes that are predicted to be correct. It can also be called the detection rate (*DR*), as follows:

$$DR = Recall = \frac{TP}{TP + FN} \tag{3}$$

*F*1-score is a comprehensive indicator of accuracy and recall, which is expressed as:

$$F1 - score = \frac{2 * precision * recall}{precision + recall} \tag{4}$$

We use the training data of the target dataset to train two traffic classification models, namely the direct training model and the DA-Transfer model, and use the test data to test the two models. Figure 4 shows the normalized confusion matrix for the classification results of the test datasets obtained by the two models.

The average classification accuracy of the direct training model for the nine classes of the dataset is 0.72, and the DA-Transfer model reaches 0.93, demonstrating the effectiveness of our method for small-sample datasets. In terms of the classification accuracy of Finalfantasy, Irat, and Poison-ivy, the classification accuracy of the direct training model is 0.54, 0.59, and 0.43, which is because the number of packages for these three types of training is less than 90, which is insufficient to support the huge amount of data required by the neural network. The DA-Transfer method shows 0.83, 0.81, and 0.84 accuracy on these three types of data due to the transfer of the feature extraction ability of the source network. It can prove the effectiveness of our method for small-sample datasets.

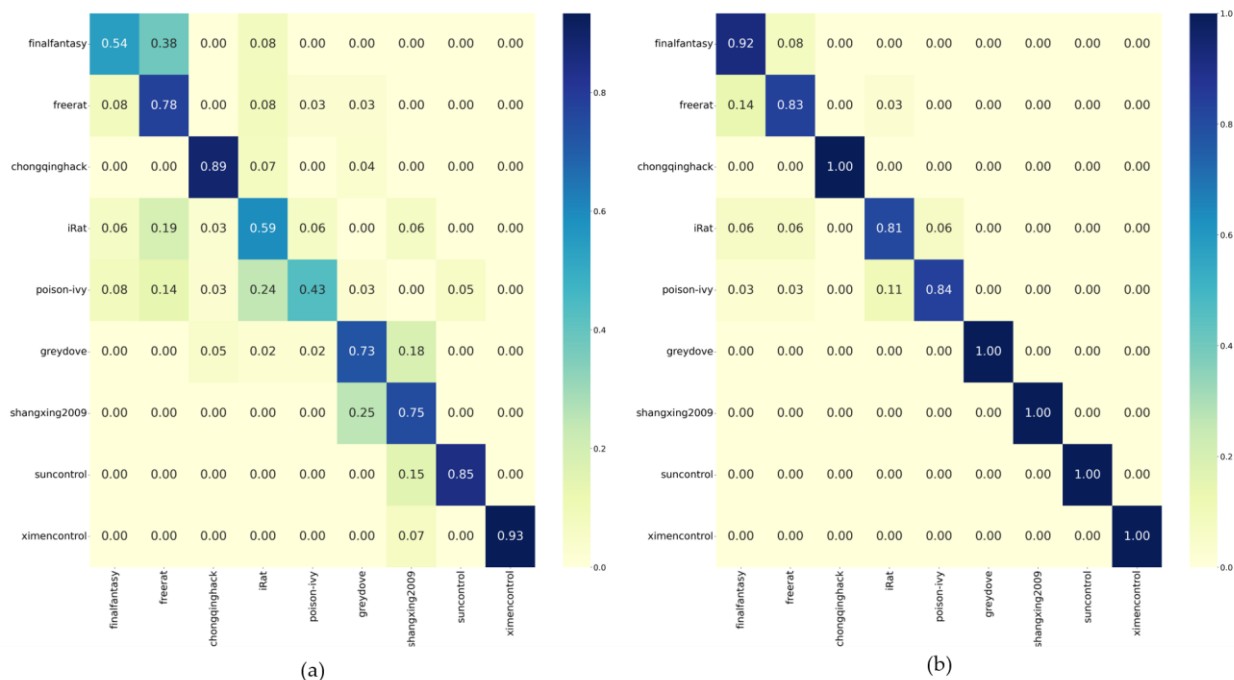

**Figure 4.** Classification accuracy of small-sample datasets. (**a**) Performance of the direct training model. (**b**) Performance of the DA-Transfer model.

Figure 5 is the loss function curve of the direct training model and the DA-Transfer model. Figure 5 shows that the direct training model begins to converge near the training round of 400, while the DA-Transfer model begins to converge near 200. Experiments show that the DA-Transfer method can transfer the knowledge trained in the source dataset to the target dataset, so as to better complete the classification task of the target dataset.

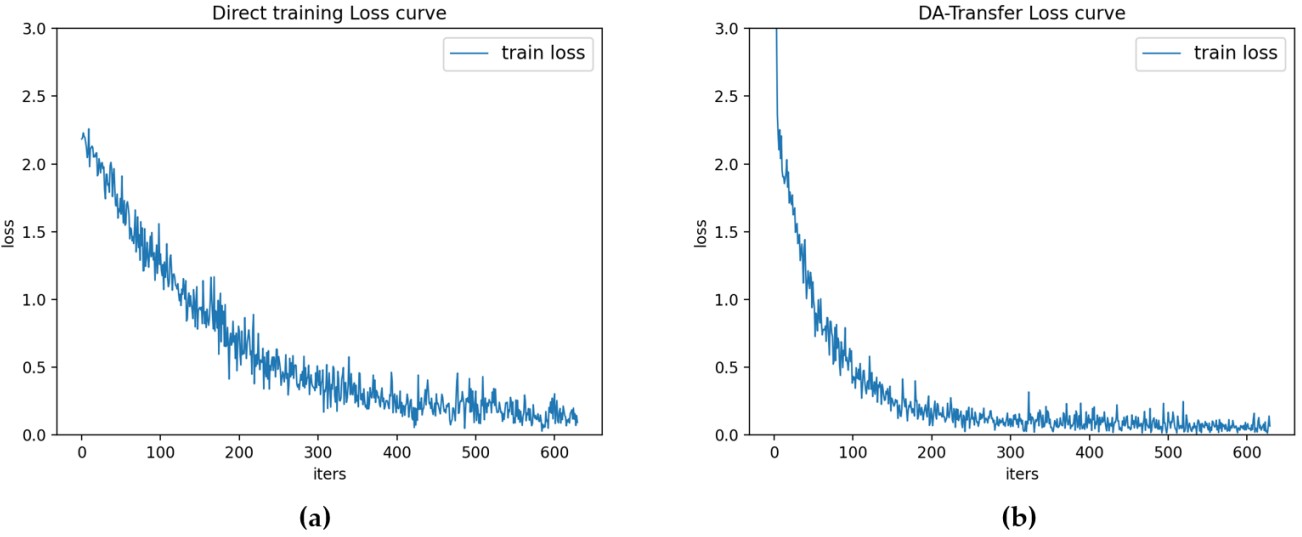

**Figure 5.** (**a**) The loss curve of the direct training model. (**b**) DA-Transfer model loss curve.

### 4.1.2. Comparison with Existing Networks

We compare the performance of six classification models, which are the direct training model, the DA-Transfer method training model, the classical traffic classification model by Deeppacket [16], CNN-LSTM [17], and the transfer classification models

proposed by Idrissi et al. [30] and Eva et al. [31] in the past two years. The accuracy, recall rate, and other measurement indicators that achieved sufficient results in the study [34,35] are used to test the model. The ROC curves and PR curves of the six models are shown in Figures 6 and 7. Precision, recall rate, and other indicators are shown in Table 5.

Figure 6 shows the ROC curves and AUC values of the six methods on each type of data in the test set. The *x* axis of the ROC curve represents the false positive rate (FPR) and the *y* axis represents the true positive rate (TPR). The curves closer to the upper left corner of the graph represent the superior performance of the classifier. It is obvious from the graph that the ROC curve of each type of DA-Transfer method is closer to the upper left corner. The AUC value of the average ROC curve represents the average accuracy of the model for nine classifications of datasets. The AUC values of the six methods are DA-Transfer (0.99), direct training (0.92), Deeppacket (0.96), CNN-LSTM (0.96), Idriss (0.93), and Eva (0.97). The results show that the DA-Transfer method is superior to other methods.

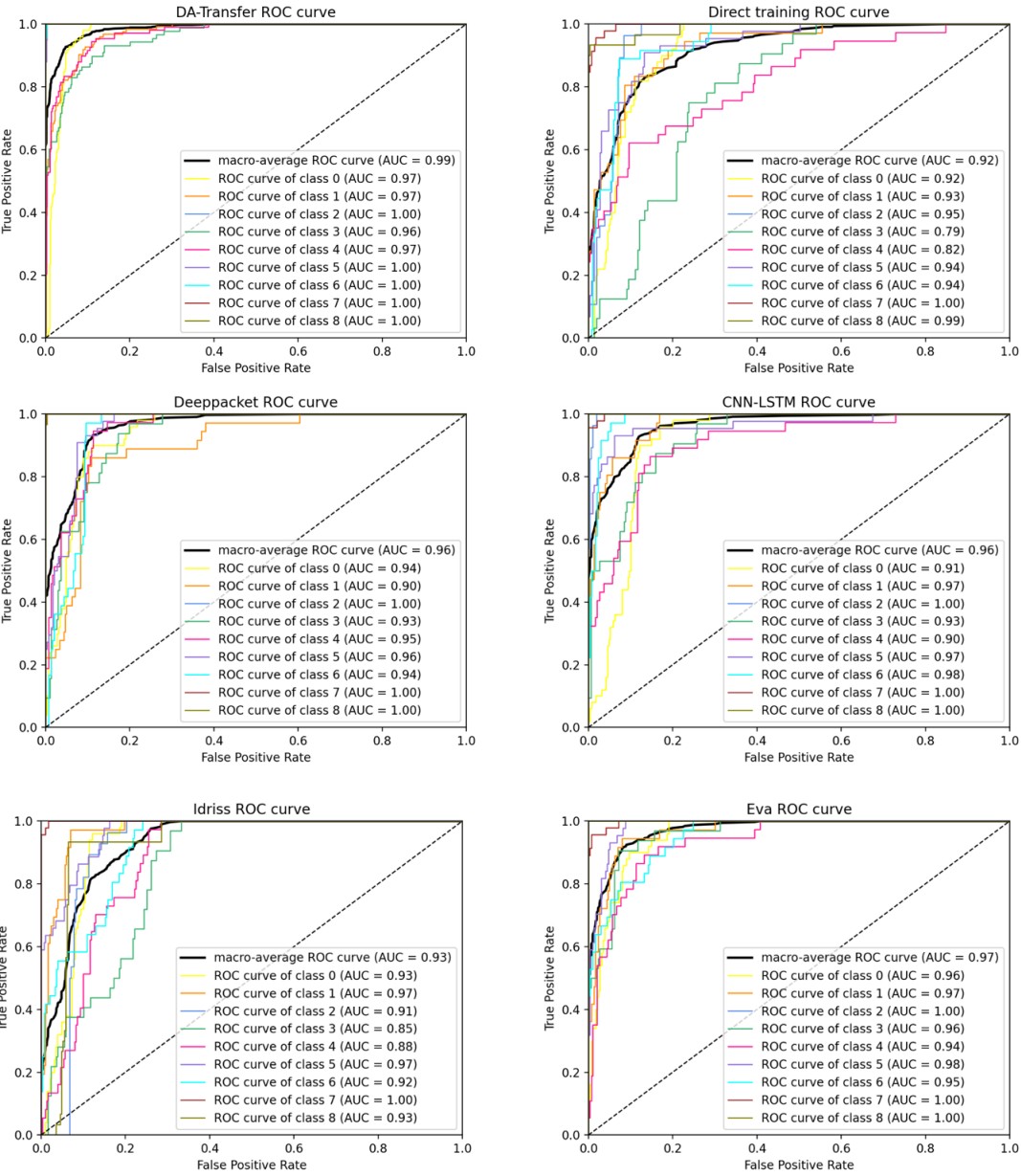

**Figure 6.** ROC curve of six traffic classification methods: DA-Transfer, direct training, Deeppacket, CNN-LSTM, Idriss, and Eva.

When the positive and negative distributions of the test samples are uneven, PR can more effectively reflect the quality of the classifier than ROC. In order to comprehensively evaluate the model, we evaluated the PR indicators for the six models. We calculate the average precision (macro-Prediction) and average recall (macro-Recall). We draw the macro-Average PR curve and calculate the mean average precision (mAP), as shown in Figure 7. When the accuracy and recall rate are high, the model performance is better, so the curve is expected to be close to the upper right corner. Figure 7 shows that the PR curve for the DA-Transfer method is closer to the upper right corner and the mAP is higher. Therefore, the average classification performance of the DA-Transfer model for the nine test set classifications is better than the other methods.

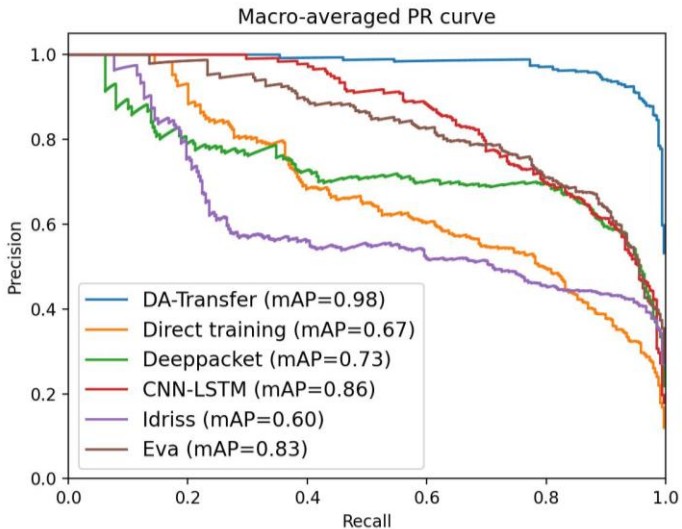

**Figure 7.** Macro-Average PR curves of six classification models.

We evaluated the prediction, recall, $F$1-score, and $t$-test metrics of six classification models, as shown in Table 5. The DA-Transfer method is superior to other methods in prediction, recall, and $F$1-score. We also conducted a 10-fold cross-validation and calculated the standard deviation to evaluate the model uncertainty. The results show that the DA-Transfer method is equal to the Eva method in model stability and superior to the other four methods. The $t$-test indicator shows significant differences between DA-Transfer and the other five methods. A $t$-test value of less than 0.05 (***) indicates a significant difference between the two models. Table 5 shows that DA-Transfer has significant differences compared with direct training, Deeppacket, CNN-LSTM, and Idriss, and has no significant difference from the Eva method. The above experimental results show that the DA-Transfer method has a slight advantage over the Eva method and is significantly better than the other four methods.

**Table 5.** Prediction, recall, $F$1-score, and $t$-test index of the six classification methods. ***means that the statistical test has a significant difference.

| Model | *Prediction* | *Recall* | *F*1-Score | *t*-Test |
|---|---|---|---|---|
| DA-Transfer | 92.8 (±1.53) | 92.83 (±0.82) | 92.81 (±0.55) | - |
| Direct training | 79.44 (±2.84) | 85.66 (±1.56) | 85.28 (±1.23) | 0.0005 (***) |
| Deeppacket | 76.42 (±1.99) | 77.90 (±1.12) | 77.15 (±0.84) | 0.0003 (***) |
| CNN-LSTM | 82.85 (±3.14) | 83.26 (±1.68) | 83.05 (±1.44) | 0.007 (***) |
| Idriss | 89.27 (±1.91) | 89.35 (±2.01) | 89.31 (±1.81) | 0.001 (***) |
| Eva | 91.86 (±1.30) | 91.54 (±1.02) | 91.70 (±0.72) | 0.19 |

4.1.3. Ablation Experiments

We set up ablation experiments on the target dataset to demonstrate the importance of each module of our proposed DA-Transfer method by testing the accuracy of four classification models, which are trained as follows: direct training of randomly initialized networks, add data adaptive module training network, training a network using model adaptation, and both add data adaptive module and model adaptive module training network.

Figure 8 shows the classification accuracy of each model for each category. More regular distribution after data adaptation helps the model to distinguish classification boundaries during training, so the classification accuracy of the model is slightly improved compared to non-adapted data.

Transferring the network that has undergone model adaptation further trains the model to extract advanced features for small-sample datasets while retaining the feature extraction capability of the source network, and its accuracy is higher than those of the first two models without transfer. The data adaptive transfer network reduces the distance between the source dataset and the target dataset, and inherits the feature extraction ability of the transfer network to achieve the best classification accuracy. The result shows that both the data adaptation module and the model transfer adaptation module of the proposed DA-Transfer method play important roles in the performance of the final model.

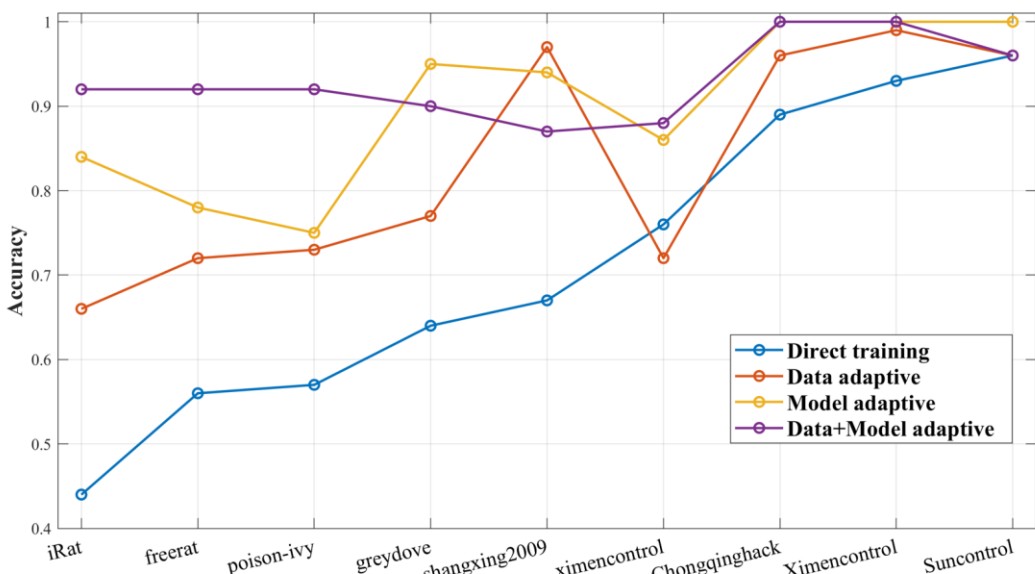

**Figure 8.** Classification accuracy of ablation experiments on nine data classes.

### 4.2. Data Adaptation Performance

We designed a data mapping network to reduce the distribution distance between target and source datasets and to verify the effectiveness of the mapping network. We introduce the t-SNE (t-distributed stochastic neighbor embedding) algorithm [36] to handle the data visualization. This algorithm is a nonlinear dimensionality reduction algorithm that can reduce the high-dimensional data into two dimensions, better displaying the distribution of the data.

Figure 9a shows the original distribution distances of the source and target datasets, where the red circles represent the data distribution of the target dataset and the blue circles represent that of the target data, from which it can be seen that the data distribution of the target dataset is scattered and differs greatly from that of the source dataset.

The scattered data distribution leads to poor classification boundaries, easily resulting in poor classification performance, while the large difference in data distribution of

the source and target datasets causes a larger adjustment of the transferred model parameters, resulting in low model training efficiency. After the data adaptation module, we use the MMD distance as the loss function to reduce the distribution distance between the source and target datasets from 6.49 to 0.16. As can be seen in Figure 9b, the distribution of the target dataset after performing data adaptation is more regular, and the distribution distance between the source and target datasets is significantly reduced.

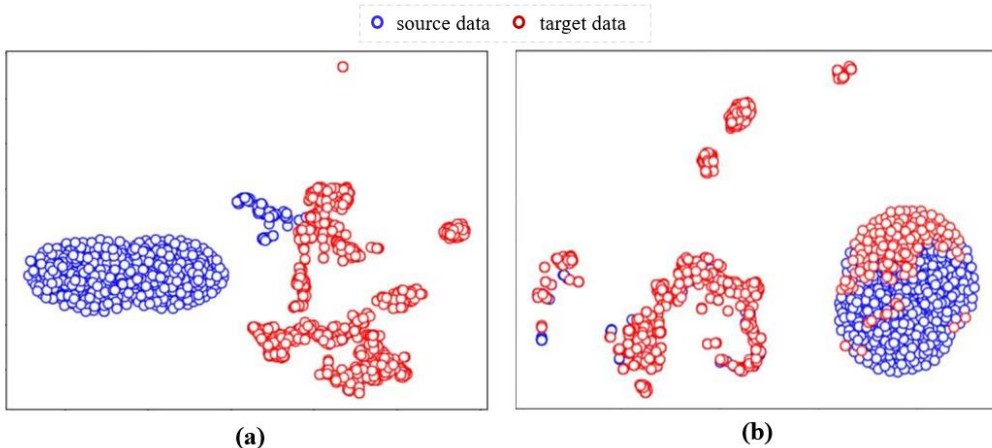

**Figure 9.** (**a**) Original data distribution of source and target datasets. (**b**) Data distribution of source and target datasets after data adaptation module.

### *4.3. Time Consumption and Effect Analysis*

In order to evaluate the efficiency of our proposed DA-Transfer method, we tested the time consumption of the traditional pre-training and fine-tuning transfer method, the DA-Transfer method, and classical Deeppacket and CNN-LSTM traffic classification methods in training 200 packets with 80% classification accuracy. Since the Idrissi and Eva transfer classification methods do not select the optimal number of frozen layers during model transfer, they freeze all convolutional layers. Therefore, the comparison of time consumption with their transfer and training is not referential. Through the comparison of model accuracy in Section 4.2, it can be found that the transfer methods of Idrissi and Eva need to be improved for model accuracy. Figure 10 shows the time consumption of the four methods, where the blue part shows the time required to determine the transfer freeze parameter (only the traditional model transfer method and DA-Transfer method have this part), the orange part shows the time required for model training, and the yellow part shows the time required for data adaptation (including the training of the mapping network and mapping the dataset, which is included in the DA-Transfer method only).

As can be seen, since the traditional transfer method requires layer-by-layer training to find the transfer freeze parameters, our proposed model transfer adaptive module is capable of adaptively recommending freeze parameters without retraining, which reduces the time spent on this part by about 42%. At the same time, since the target dataset mapped by the data adaptive module has a smaller amplitude for the fine-tuning model, the model fine-tuning time represented by the orange part is shortened by about 51% compared to the traditional transfer method. The classical Deeppacket and CNN-LSTM traffic classification methods contain quantity parameters that need to be trained. Therefore, the model requires more training rounds, and accordingly, the training time increases. Therefore, even with the addition of the time consumption of the data adaptation module (yellow part in the figure), which is not available in the other three methods, the total time consumption of the DA-Transfer method training network is 20.01% higher than that of traditional model transfer methods, and is efficient in traffic classification.

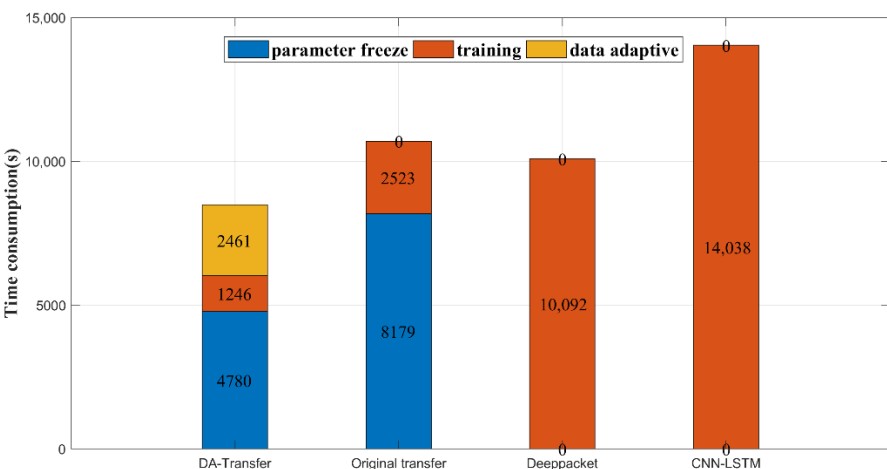

**Figure 10.** Time consumption of four classification methods.

### 4.4. Model Transfer Improves Efficiency

The problem of low efficiency and reliance on manual experience when freezing parameters of traditional screening transfer networks needs to be solved. In this paper, a model transfer adaptive module is designed, which converts the traditional task of finding the number of shared parameter layers into a binary classification problem in the source domain and target domain. This method has no special requirement for the source and target domains. Therefore, we believe that this method is generalized.

Figure 11 shows the time consumption of our proposed method compared to the traversal search method in recommending frozen parameters, and the effect of freezing parameters of different layers of the network on the classification accuracy. Compared with the traversal search method, the model adaptation module does not need to retrain the parameters other than the frozen layer. Since it converts the problem into a binary classification problem, it also reduces huge parameters in the linear layer compared with the multi-classification problem, shortening the time to recommend the freezing of parameters.

The line in Figure 11 shows the performance of the trained network in the classification accuracy of the small-sample datasets when freezing parameters of different layers. Freezing less than two layers makes it difficult to train the huge number of remaining network parameters for small-sample datasets, which affects the classification accuracy. As the number of frozen layers increases, the number of network layer parameters that need to be data-driven decreases, and the network classification accuracy gradually increases. When the number of frozen layers exceeds the recommended number of layers by two, the parameters of the frozen network layers may extract the high-level features of the source dataset instead of the common features of the two datasets, which makes the remaining parameters insufficient to extract the high-level features of the target dataset. This causes the final extracted features to fail in completing the classification task, and the network classification accuracy gradually decreases as the number of frozen layers increases.

We presume that the appropriate network freezing layer parameters are crucial for classification accuracy. The model transfer adaptive module proposed in this paper has improved both the accuracy and efficiency of the model.

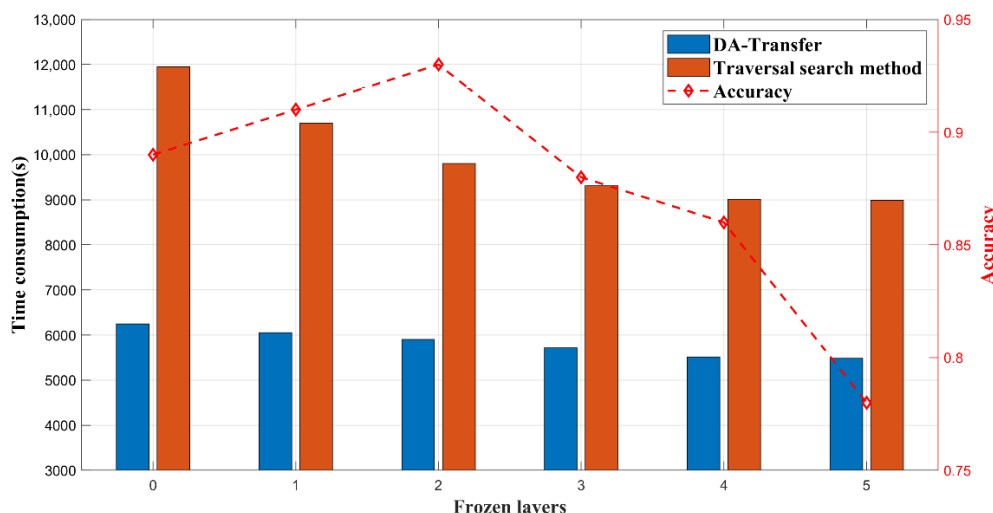

**Figure 11.** The freezing parameter recommendation module improves the accuracy and efficiency of the network model.

## 5. Discussion

### 5.1. Model Analysis

In order to resolve the contradiction between the small-sample datasets and the huge number of deep learning parameters, we propose a small-sample malicious traffic classification method based on deep transfer learning, which adds model transfer adaptation and data adaptation modules to the traditional transfer learning method.

The traditional method relies on the experience of the network model designers, which is not suitable for the current traffic data iteration speed and does not have generalization ability. We propose an adaptive network transfer module that enables the model to adaptively freeze parameters of the transfer model when performing transfer based on the variability between the source and target datasets.

Meanwhile, since small-sample datasets have limited ability to fine-tune the transfer model, and as the variability between the source and target datasets has a significant impact on the accuracy of the fine-tuned model, we designed the data adaptation module in the traffic classification framework. The traditional processes of data adaptation are often based on feature-level mapping, and such methods may lose the high-level features of the target dataset, while the small-sample dataset needs to be mined for features that can be distinguished from each other among different classes in a small number of samples. Therefore, we designed a neural network-based data adaptation module, which works on the original data and enables the target dataset close to the source dataset. After the data adaptation module, the target dataset needs less fine-tuning for the parameters of the transfer network, and it is better adapted to the transfer network, improving the classification accuracy of the model for small-sample datasets.

We make a fusion of the model and data adaptation modules to jointly serve the model transfer for small-sample datasets. This approach reduces the difficulty in transferring model fitness and transfer efficiency. The fusion method significantly improves the test classification performance, with an average accuracy of 0.93 and a 20.01% improvement in the training efficiency.

*5.2. Limitations and Future Research*

In this paper, we explored the task of classifying small samples of network traffic. The proposed DA-Transfer method demonstrates better performance than the conventional methods. Moreover, the existing methods have some limitations that need attention, and further improvements are planned for future research.

Firstly, the DA-Transfer method based on the deep network module is not interpretable enough to condense the experience on traffic features. The coding and decoding of packet structures by neural networks are the basis of traffic classification, especially the realistic meaning corresponding to the characterization features. As a data transmission structure with context semantics, the traffic packet has rich meanings in the paragraph information of a single data segment and the related information in the previous and later texts. The DA-Transfer extracts low-level and high-level network traffic features at the bottom and top levels, respectively. These features are difficult to summarize in the current interpretation. The end-to-end learning method improves the training effect, but visualization is needed to understand the representation mechanism of features.

Secondly, the recommendation algorithm with frozen layers in the transfer learning improves the training efficiency, and more network parameters can apply a similar tuning strategy. At present, the deep network realizes automatic feature extraction, meta-learning, and box self-search. Other methods provide suggestions on the network structure. The DA-Transfer method has achieved considerable benefits in the frozen layer. Future research can further expand the scope of parameter optimization and provide a combination strategy for multiple parameters of the overall network design.

Finally, the real-time network traffic classification needs further optimization, including the real-time performance of the algorithm output and fast iteration of the real network environment. The real-time performance of the algorithm requires the simplicity and efficiency of the method. The DA-Transfer method retains the original large-scale network structure and parameter quantity, which requires certain computing resources and storage space. The huge network is difficult to apply on portable devices, and the parameter redundancy occupies a large amount of device storage. In future research, the model size can be simplified by the lightweight method of knowledge distillation, or pruning to reduce the equipment operation basis of the algorithm and expand its application scenarios. Upon facing the problem of rapid iteration in the network environment, the classification model should have the ability to capture and identify new application traffic in real time. The DA-Transfer method needs to adjust with the changes in the network environment, open up new category space, and expand the classification ability of new samples through a class incremental method, so as to face the updating and iteration of practical applications.

## 6. Conclusions

We propose a malicious traffic classification method for small-sample datasets. The method combines both data adaptation and model transfer adaptation components. In the data adaptation, we designed a mapper based on neural networks to reduce the distribution distance between the mapped target and source datasets, and the transfer classification model can be better adapted to the target dataset. In addition, in the model transfer, we propose an adaptive parameter freezing method, which can accurately and efficiently determine the number of network layers that can freeze parameters during model transfer. Through the method of data adaptive and model adaptive fusions, we finally trained a classification network that demonstrates suitable classification accuracy and efficiency on small-sample datasets. This paper can provide a new perspective for deep network transfer tasks as well as small-sample classification tasks.

**Author Contributions:** R.W. is mainly responsible for research design, data analysis, and manuscript writing for this study. J.F. is mainly responsible for data collection and manuscript editing. M.Z. is mainly responsible for research design. R.Z. is mainly responsible for research design. M.G.

is mainly responsible for data collection and production of charts. X.L. and Z.Q. are mainly experiment design and preparation. All authors have read and agreed to the published version of the manuscript.

**Funding:** This work was supported by the National Key Research and Development Project of China (2019QY1302).

**Institutional Review Board Statement:** Not applicable.

**Informed Consent Statement:** Not applicable.

**Data Availability Statement:** Publicly available datasets were analyzed in this study. These data can be found at: https://www.unb.ca/cic/datasets/vpn.html (Accessed 5 June 2020).

**Conflicts of Interest:** The authors declare no conflict of interest.

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
