# Peer review of "DA-Transfer: A Transfer Method for Malicious Network Traffic Classification with Small Sample Problem"

_electronics, doi:10.3390/electronics11213577_

Round 1
Reviewer 1 Report
In this study, the authors proposed a transfer method for malicious network traffic classification. The performance was promising, however, some major points should be addressed as follows:
1. In my opinion, the authors should improve the methodology part in terms of model implementation. The current version only lists basic information regarding the model without any detailed implementation (i.e., hyperparameters, weights, layers, etc.).
2. How did the authors tune the optimal hyperparameters of the models?
3. Uncertainties of models should be reported.
4. When comparing the predictive performance among methods/models, the authors should conduct some statistical tests to see significant differences.
5. ROC and PR curves should be provided.
6. The authors must compare the performance to other SOTA models in this field.
7. Measurement metrics (i.e., recall, precision, etc.) are well-known and have been used in previous studies such as PMID: 34989149, PMID: 34502160. Thus, the authors are suggested to refer to more works in this description to attract a broader readership.
8. References are weak, need to be improved.
9. Quality of figures should be improved.
10. Source codes should be provided for replicating the study.
Reviewer 2 Report
Dear Authors,
Thank you for your good paper. Some concerns are given as follows:
· There are grammatical and punctuation errors in the paper. The authors require a native speaker to proofread. The authors can use the professional version of the Grammarly system.
· The authors should update the review to 2022.
· The resolution of Figure 4 needs to be increased to 200 dpi.
· Authors should specify exactly what they mean by the original method in Figure 6 and mention the name of this method in the related text.
· The authors should assign Receiver Operating Characteristics Curve in the results section and put the AUC values of the methods.
Round 2
Reviewer 1 Report
My previous comments have been addressed.
Reviewer 2 Report
Dear Authors,
Thank you for addressing my concerns.